# Infection of Human Endothelial Cells with Lassa Virus Induces Early but Transient Activation and Low Type I IFN Response Compared to the Closely-Related Nonpathogenic Mopeia Virus

**DOI:** 10.3390/v14030652

**Published:** 2022-03-21

**Authors:** Othmann Merabet, Natalia Pietrosemoli, Emeline Perthame, Jean Armengaud, Jean-Charles Gaillard, Virginie Borges-Cardoso, Maïlys Daniau, Catherine Legras-Lachuer, Xavier Carnec, Sylvain Baize

**Affiliations:** 1Unité de Biologie des Infections Virales Emergentes, Institut Pasteur, 69007 Lyon, France; othmann.merabet@gmail.com (O.M.); virginie.borges-cardoso@pasteur.fr (V.B.-C.); xavier.carnec@pasteur.fr (X.C.); 2Centre International de Recherche en Infectiologie (CIRI), Université de Lyon, INSERM U1111, Ecole Normale Supérieure de Lyon, Université Lyon 1, CNRS, UMR5308, 69007 Lyon, France; 3Bioinformatics and Biostatistics Hub, Institut Pasteur, Université de Paris, 75015 Paris, France; natalia.pietrosemoli@pasteur.fr (N.P.); emeline.perthame@pasteur.fr (E.P.); 4Laboratoire Innovations Technologiques pour la Détection et le Diagnostic (LI2D), Service de Pharmacologie et Immunoanalyse (SPI), Commissariat à l’Energie Atomique, 30200 Bagnols-sur-Cèze, France; jean.armengaud@cea.fr (J.A.); jean-charles.gaillard@cea.fr (J.-C.G.); 5ViroScan3D SAS, 01600 Trévoux, France; mailys.daniau@viroscan3d.com (M.D.); catherine.lachuer@viroscan3d.com (C.L.-L.)

**Keywords:** Lassa virus, endothelial cells, viral hemorrhagic fevers, Mopeia virus, pathogenicity

## Abstract

Lassa virus (LASV), an Old World arenavirus, is responsible for hemorrhagic fevers in western Africa. The privileged tropism of LASV for endothelial cells combined with a dysregulated inflammatory response are the main cause of the increase in vascular permeability observed during the disease. Mopeia virus (MOPV) is another arenavirus closely related to LASV but nonpathogenic for non-human primates (NHPs) and has never been described in humans. MOPV is more immunogenic than LASV in NHPs and in vitro in human immune cell models, with more intense type I IFN and adaptive cellular responses. Here, we compared the transcriptomic and proteomic responses of human umbilical vein endothelial cells (HUVECs) to infection with the two viruses to further decipher the mechanisms involved in their differences in immunogenicity and pathogenicity. Both viruses replicated durably and efficiently in HUVECs, but the responses they induced were strikingly different. Modest activation was observed at an early stage of LASV infection and then rapidly shut down. By contrast, MOPV induced a late but more intense response, characterized by the expression of genes and proteins mainly associated with the type I IFN response and antigen processing/presentation. Such a response is consistent with the higher immunogenicity of MOPV relative to LASV, whereas the lack of an innate response induced in HUVECs by LASV is consistent with its uncontrolled systemic dissemination through the vascular endothelium.

## 1. Introduction

Lassa hemorrhagic fever (LF) is responsible for between 5000 and 6000 deaths every year. The etiological agent of LF is Lassa virus (LASV), a member of the Arenaviridae family isolated in 1969 in Nigeria [1]. This family consists of four genera, including *Mammarenavirus*. This genus can be divided into two complexes, depending on their antigenic properties. The New World arenavirus complex contains highly pathogenic viruses, such as the Junin, Machupo, Sabiá, Chapare, Guanarito, and Whitewater Arroyo viruses [2,3,4,5]. The Old World arenavirus complex contains lymphocytic choriomeningitis virus (LCMV), which can lead to fetal infection and severe sequelae [6]. The hemorrhagic fever associated Lujo virus (isolated in 2008) and LASV are also members of this complex [7].

The genome of these viruses is composed of two single-stranded RNA molecules, a small segment (S) and a large segment (L) [8]. Each segment carries two open reading frames (ORFs) in opposite orientations. The S segment codes for the nucleoprotein (NP) and the glycoprotein complex (GPC). The L segment codes for the RNA dependent RNA polymerase (Lpol) and the matrix protein (Z). Viral infectious particles are formed by a ribonucleoprotein (RNP), comprised of at least one of each segment associated with NPs and the Lpol. This RNP is the minimal infective unit and interacts with the Z protein, which is found inside the viral membrane. The viral membrane is derived from the cellular membrane and is covered by glycoprotein trimers.

The main reservoir of LASV is the peri-domestic rodent *Mastomys natalensis* [9]. Human transmission occurs through direct or indirect contact with animal body fluids [10,11]. Infection is also possible by the inhalation of virions from urine or feces or mucocutaneous transmission through contaminated bushmeat/food [12]. Although the geographic distribution of *Mastomys natalensis* is large, LASV is solely endemic in western Africa (mainly in Nigeria, Liberia, Sierra Leone, and Guinea). Human-to-human transmission is also possible after contact with contaminated patients [13]. Outbreaks are frequently reported to the World Health Organization (WHO), and LF is the most highly imported hemorrhagic fever to occidental countries [14]. Currently, there is no licensed vaccine or treatment available. Ribavirin is recommended by the WHO to treat LF, but its efficiency appears to be very limited [15]. For these reasons, LASV is a major public health problem.

Mopeia virus (MOPV) is an arenavirus closely related to LASV [16]. Comparative analysis has shown 75% identity for the amino-acid sequence and they share the same main reservoir [17]. No MOPV-related human infections have yet been reported, and MOPV is nonpathogenic in non-human primates (NHPs). Moreover, a previous MOPV infection protects primates from lethal LASV challenge [18]. The difference in pathogenicity appears to be associated with different immunogenicity of the two viruses, in particular, in terms of the IFN response. Our previous results have shown distinct responses of macrophages and dendritic cells to infection by MOPV or LASV [19,20,21].

The two main routes of human infection by LASV are the respiratory tract and mucous membranes. Antigen-presenting cells are the first target of LASV [19,22,23], and the virus is subsequently transported by these cells to the secondary lymphoid organs (spleen and lymph nodes) [24]. The viruses disseminate to almost all organs through the blood and lymph. Endothelial cells (ECs) are massively infected during LF. Only minor vascular lesions can be observed, in contrast to massive endothelial dysfunction [25]. ECs are activated by infection and diapedesis and release high amounts of intercellular adhesion molecule (ICAM), P-selectin, and the endothelial protein C receptor (EPCR) [26]. Platelet aggregation is also reduced, leading to the dysfunction of hemostasis.

We further characterized the differences between LASV and MOPV in target cells highly relevant to LASV-associated pathology by analyzing the transcriptomic and proteomic responses of primary ECs to LASV or MOPV infection using RNA sequencing and mass spectrometry. We confirmed that both viruses replicate efficiently and durably in human umbilical vein ECs (HUVECs). However, they demonstrated strikingly different responses depending on the virus. LASV induced moderate and transient activation early after infection, whereas a more intense response evocative of the IFN response and antigen processing/presentation was induced at a later stage by MOPV. These data provide insights on the endothelial dysfunction observed during LF.

## 2. Materials and Methods

### 2.1. Viruses and Cells

Strain AN21366 (GenBank accession number JN561684 and JN561685) was used for MOPV infection, and Strain AV (GenBank accession number FR832711 and FR832710) for LASV infection. HUVECs were obtained from three Caucasian donors (Promocell, Heidelberg, Germany, lot number 434Z003, 428Z011.3, and 433Z026). Cells were grown using the EC Growth Medium 2 kit (Promocell, Heidelberg, Germany) according to the manufacturer’s instructions. HUVECs were infected or not with MOPV or LASV at an MOI of 1 for 1 h. A positive control for activation of the cells was included, consisting of treatment with TNFα at a concentration of 20 ng/mL for 18 h. Culture supernatants and cells were collected 18 or 48 h post infection. All experiments performed with LASV were done in the Jean Mérieux-INSERM BSL4 laboratory (Lyon, France). Experiments performed with MOPV were done in the same conditions to ensure reproducibility.

### 2.2. Titration of Viruses

Vero E6 cells were infected with sequential dilutions of virus-containing supernatant and incubated at 37 °C in 5% CO_2_ for seven days with carboxy-methyl-cellulose (1.6%) (BDH Laboratory Supplies, Poole, UK) in Dulbecco Modified Eagle’s Medium (DMEM, Life Technologies—Thermo Fisher Scientific, Cailloux-sur-Fontaines, France) supplemented with 1.5% fetal bovine serum (FBS, Eurobio Scientific, Les Ulis, France) and 0.5% penicillin–streptomycin (PS, Life Technologies—Thermo Fisher Scientific, Cailloux-sur-Fontaines, France). Cells were fixed with 4% formaldehyde (Sigma Aldrich, Saint-Quentin-Fallavier, France) diluted in phosphate buffered saline (PBS, Sigma Aldrich, Saint-Quentin-Fallavier, France) and permeabilized with Triton X-100 (1/1000 in PBS, Sigma Aldrich, Saint-Quentin-Fallavier, France). Infectious foci were detected by incubation with monoclonal antibodies (mAbs) directed against MOPV or LASV (mAbs L52-54-6A, L53-237-5, and YQB06-AE05, generously provided by P. Jahrling, USAMRIID, Fort Detrick, MD, USA), followed by PA-conjugated goat polyclonal anti-mouse IgG (Sigma Aldrich, Saint-Quentin-Fallavier, France).

### 2.3. Measurement of the Viral Load

Viral loads were measured by reverse transcription real-time polymerase chain reaction (RT-qPCR). For MOPV, RT-qPCR was performed using the SensiFAST Probe No-ROX One-Step kit (Bioline, London, UK) with the primers 5′-GTCAAGCGTTCTTTGGGAATG-3′ and 5′-TCCAGAAAGACATAGTTTGTAGAGG-3′ and probe: 5′-FAM-TTCCTTTCCCCTGGCGTGTCA-BHQ1-3′. For LASV, RT-qPCR was performed using the Eurobiogreen Lo-Rox qPCR mix kit (Eurobio Scientific, Les Ulis, France) with the primers 5′-CTCTCACCCGGAGTATCT-3′ and 5′-CCTCAATCAATGGATGGC-3′. Amplification and reading were performed using a LightCycler 480 II device (Roche Diagnostics, Meylan, France).

### 2.4. Protein Extraction

HUVECs were infected or not with MOPV or LASV at an MOI of 1 for 1 h. Culture supernatants and cells from three different donors were collected 18 or 48 h post infection. Immunoglobulins were depleted from the supernatant by antibody purification using a protein G kit (Thermo Fisher Scientific, Cailloux-sur-Fontaines, France) according to the manufacturer’s instructions. Trichloroacetic acid (Sigma Aldrich, Saint-Quentin-Fallavier, France) was added to a final concentration (Vol/Vol) of 10%. After centrifugation at 2000× *g* for 15 min, the pellets were reconstituted with NuPAGE LDS Sample Buffer (Invitrogen, Thermo Fisher Scientific, Cailloux-sur-Fontaines, France) supplemented with BME (β-mercaptoethanol) at a final concentration of 2.5%. Cells were also collected after trypsinization, pelleted, and lysed with NuPAGE LDS Sample Buffer supplemented with BME. Samples were heated to 60 °C for 60 min for viral inactivation.

### 2.5. Trypsin Proteolysis and Tandem Mass Spectrometry

For each sample, obtained from three different cell donors, 30 μL was subjected to short (5 min) denaturing electrophoresis on a NuPAGE 4–12% gradient gel in MES SDS running buffer (50 mM MES 2-N-morpholino ethane sulfonic acid), 50 mM Tris Base, 0.1% SDS, 1 mM EDTA, pH 7.3). After brief staining with SimplyBlue SafeStain (Thermo Fisher Scientific, Cailloux-sur-Fontaines, France) and de-staining overnight in MilliQ water, each proteome was extracted as a single polyacrylamide band. Each sample was proteolyzed with trypsin Gold (Promega, Charbonnières-les-Bains, France) in the presence of ProteaseMax detergent (Promega, Charbonnières-les-Bains, France). From the 50 µL resulting peptide mixture, 4 µL was injected into a nanoscale C18 PepMap100 capillary column (3 µm, 100 Å, 75 µm id × 50 cm, LC Packings, Conquer Scientific, Poway, CA, USA) and resolved with a 120-min gradient of CH_3_CN (3.2–20% over 100 min followed by 20–32% over 20 min) and 0.1% formic acid at a flow rate of 0.2 µL/min. Data-dependent acquisition analysis of the peptides eluting from the column was performed using a Q-Exactive HF mass spectrometer (Thermo Fisher Scientific, Cailloux-sur-Fontaines, France). Full scans of peptide ions with a 2+ or 3+ charge were acquired at a resolution of 60,000 from *m*/*z* 350 to 1500 with a dynamic exclusion of 10 s. Each MS scan was followed by high-energy collisional dissociation and MS/MS scans at a resolution of 15,000 on the 20 most abundant precursor ions.

### 2.6. Protein Quantification in Cell Supernatants

Concentrations of ADAMTS13, angiopoietin, angiostatin, BMP-9, BNP, Chitinase3 like1, CK-MB, CXCL10, CXCL16, CXCL6, cystatin C, D-dimer, EGF, endocan-1, endoglin, endothelin-1, FABP3/4, FGF-1/2, follistatin, G-CSF, GDF-15, Granzyme B, HB-EGF, IL-12p70, IL-17, IL-18, IL-1α, IL-1β, IL-6, IL-8, leptin, LIGHT, lipocalin-2, MPO, myoglobin, NTproBNP, OSM, osteopontin, PDGF-AB/BB, PLGF, procalcitonin, P-selectin, SAA, sAXL, sc-kit, sEGFR, sE-selectin, sHER2/3, sHGFR, s-ICAM1, sIL6-RA, s-neurophilin1, sPECAM1, sTIE-2, suPAR, sVCAM1, sVEGFR1/2/3, tenascin C, thrombospondin-2, TNFα, TNF-RI, TREM1, troponin1, and VEGF-A/C/D were quantified in cell supernatants using a bead-based immunofluorescence assay (Luminex MAGPIX^®^ system, Merck Millipore, Burlington, MA, USA) and multiplex cytokine reagents (Merck Millipore, Burlington, MA, USA). The sensitivity of the standards ranged from 2 to 32,000 pg/mL.

### 2.7. RNA Analysis

Cellular RNA was extracted using the RNeasy Mini kit (QIAgen, Hilden, Germany) following the manufacturer’s instructions and treated with the DNase I Ambion kit (Thermo Fisher Scientific, Cailloux-sur-Fontaines, France). RNA sequencing was performed using the NEXTFLEX Rapid Directional RNA-seq kit (Bioo Scientific, Austin, TX, USA).

### 2.8. Proteomics Data Analysis

Separate analyses were performed for LASV- and MOPV-infected samples and their corresponding controls and the proteins identified and quantified using MaxQuant software (Max-Planck-Institute of Biochemistry, Martinsried, Germany) [27]. MOPV-infected samples were matched for protein group identification against the MS20-043_Swissprot-human_4-prot-Mopeia_2020-09-08 database (20,363 sequences), corresponding to the Swissprot human database (8 September 2020 download) (Swiss Institute of Bioinformatics, Lausanne, Switzerland) complemented with the proteins from MOPV, LASV-infected samples against the MS20-043_Swissprot-human_4-prot-Lassa_2020-09-08 database (20,363 sequences), corresponding to the Swissprot human database (8 September 2020 download) complemented with the proteins from LASV. Only human protein groups identified with at least two peptides were considered for the analysis, originally identifying 3940 and 3902 proteins, respectively, with 3777 proteins present in both runs. Statistical analyses were performed with R software (v 4.1.1) (R Foundation for Statistical Computing, Vienna, Austria). The dataset was corrected for the batch effect introduced by the technical bias of having one separate run for each virus by fitting a linear model to the data, including both batches and regular treatments, and then removing the component due to the batch effect (limma package v 3.48.3) (Bioconductor) [28]. Missing LFQ values were imputed using the iterative regularized PCA method (missMDA package v 1.18) (Factominer) [29]. The differential analysis was performed using the limma package with one linear model adjusted for the effect of the virus and the timepoint, considering the replicate effect for each protein. For statistical analysis and to assess differential expression, limma uses an empirical Bayes method to moderate the standard errors of the estimated log-fold changes. Pairwise comparisons between groups were performed and *p*–values adjusted using the Benjamini–Hochberg multiple testing correction method [30], which controls for the expected false discovery rate (FDR) below the specified value (FDR ≤ 0.05).

### 2.9. Functional Enrichment Analysis of Proteomics and Transcriptomics Datasets

Functional enrichment analysis was performed using the Camera6 (competitive gene set test accounting for inter-gene correlation) method to explore gene-set enrichment and pathway analysis based on the limma R package (Bioconductor) [31]. Functional annotation of the proteins/genes was obtained using the Hallmark gene sets and Kegg biological pathway collections from the MSigDB database (GSEA, UC San Diego, CA, USA) [32].

### 2.10. RNA Isolation and Sequencing

RNA was isolated from HUVECs and the samples quantified using the Quantifluor RNA system (Promega, Charbonnières-les-Bains, France). RNA sample quality control was performed using the Standard Sensitivity RNA Kit on a Fragment Analyzer (Advanced Analytical Technologies Inc., Ankeny, IA, USA) and the Pico RNA Kit on a Bioanalyzer (Agilent Technologies, Santa Clara, CA, USA). All RNA samples were validated before proceeding to library preparation, with an RNA quality number (RQN) > 7 for all samples and sufficient concentrations/quantities. Libraries were prepared using the NextFlex Rapid Directional library preparation kit for NextSeq (BiooScientific, Austin, TX, USA). Sequencing of 33 samples was performed on an Illumina NextSeq platform to generate single end 75-bp reads. The read quality of each run was assessed according to Illumina thresholds, for which between 92 and 95% of PF read bases were higher than Q30.

### 2.11. RNA-Seq Mapping and Quantification

After trimming of the adaptors using Cutadapt, v 2.4 (Dortmund University, Dortmund, Germany) [33], high-quality samples averaging 72.3 million reads were obtained. Sequencing quality was assessed for each sample (before and after mapping) using MultiQC9 v 1.6 (Stockholm University, Stockholm, Sweden) [34]. Reads were aligned on human genome hg19 using Star v STAR_2.5.3a_modified. Read summarization and annotations were performed using FeatureCounts v 1.5.3 (Bioconductor), requiring both read ends to map.

### 2.12. RNA-Seq Statistical Analysis

Gene expression profiles were analyzed using R software (v 3.6.1) (R Foundation for Statistical Computing, Vienna, Austria) and several Bioconductor packages [35], including DESeq2 v 1.26.0 (Bioconductor) [36] and SARTools v 1.7.0 (Institut Pasteur, Paris, France) [37]. The statistical analysis included (i) data description and quality control, (ii) data normalization and exploration, (iii) and testing for differential expression for each gene between the time points and concentrations. The dataset consisted of 33 samples: LASV- or MOPV-infected cells, as well as uninfected control cells, each with three replicates. Samples were filtered for duplicated mRNA, keeping that with the highest variability in expression among the samples, resulting in 20,800 genes identified for each condition. Data were normalized according to the DESeq2 model and package. Normalized read counts were obtained by dividing the raw read counts by the scaling factor associated with the sample they belonged to (parameter locfunc = “median”). The variability of the data was explored by performing hierarchical clustering and principal component analysis (PCA) of the entire sample set after the counts were transformed using a variance stabilizing transformation. Hierarchical clustering was performed using the Euclidian distance and the Ward criterion for agglomeration. PCA was performed using the DESeq2 R package to explore the structure and clustering of the samples. Differential analysis was performed to identify genes showing significantly different expression between each timepoint for the two viruses using the R packages DESeq2 and SARTools. The strategy consisted of fitting one linear model per gene using a design of one factor (condition + timepoint) to estimate the coefficients (log2FC) and corresponding *p*-value. Raw *p*-values were corrected for multiple-testing using the Benjamini–Hochberg method. Genes with an adjusted *p*-value < 0.05 were considered to be differentially expressed (DE) for the given pairwise comparison.

### 2.13. Statistical Analysis

The Wald Test (in R with the packages R DESeq² and SARTools) was performed to identify genes or proteins significantly under or over expressed (*p* < 0.05). A Wilcoxon test was performed to analyze gene expression in each pathway. Mean comparisons between groups was performed using a *t*-test with Welch’s correction. Growth kinetics were compared by two-way ANOVA followed by Sidak’s multiple comparisons test.

## 3. Results

### 3.1. Lassa and Mopeia Viruses Productively Infect HUVECs

We used HUVECs as a model of primary human ECs to determine whether ECs are privileged targets for LASV and MOPV. Infection of these cells with either virus led to the release of substantial amounts of viral particles in supernatants in as little as 24 h after infection and for up to four days (Figure 1A). The viral titers were higher for LASV than MOPV 24 and 48 h after infection. However, we measured an elevated ratio of the viral RNA load in viral particle titers of approximately 1:1000 relative to the viral RNA copy numbers detected in culture supernatants (Figure 1B).

### 3.2. Lassa and Mopeia Viruses Induce Different Transcriptomic Profiles in HUVECs

We analyzed the transcriptomic profile of HUVECs by RNA sequencing (RNAseq) 6, 24, and 48 h after infection and observed a modification during the kinetics of the cultures, including in the mock condition (Figure 2A). LASV and MOPV induced striking changes in mRNA expression by HUVECs. Only a few genes were differentially expressed (DE) from 6 h after infection between cells infected with the two viruses, but at this timepoint, seven genes belonging to the IFN-stimulated gene family were strongly upregulated after LASV infection versus MOPV infection (Figure 2B). The genes were 2′5′-oligoadenylate synthetase 1 (OAS1), epithelial–stromal interaction 1 (EPSTI1), interferon-alpha inducible proteins 6 (IFI6) and 44L (IFI44L), interferon-induced proteins with tetratricopeptide repeats 1 (IFIT1) and 2 (IFIT2), and helicase with zinc finger 2 (HELZ2). The number of DE genes increased by 24 h after infection compared to the mock condition and, again, more genes were expressed after LASV than MOPV infection. Finally, we observed the opposite pattern 48 h after infection. Indeed, although the number of DE genes did not increase after 24 h after LASV infection, HUVECs infected with MOPV upregulated the expression of a large number of genes relative to the mock (5437) and LASV (3996) conditions. We analyzed the pathways corresponding to the transcriptomic profiles (Figure 2C,D). Relative to the mock condition, LASV infection activated the following pathways: mTOR signaling, JAK-STAT signaling, pentose phosphate, apoptosis, NOD-like (at 24 h) and RIG-I-like receptor signaling (Figure 2C), TGFβ signaling, oxidative phosphorylation, protein secretion (at 6 h), inflammatory response, apical junction, and TNFα signaling via NFκB (Figure 2D). A number of pathways was downregulated after LASV infection: chemokine signaling, endocytosis, ribosome, MAPK signaling, adherens junctions (at 6 h), antigen processing/presentation, cell adhesion molecules (Figure 2C), protein secretion (at 24 h), IL-2 STAT5 signaling, IFNα and γ responses, IL-6 JAK-STAT3 signaling, and complement (Figure 2D). MOPV infection activated the following pathways relative to the mock condition: chemokine signaling, oxidative phosphorylation, purine metabolism, cell-adhesion molecules, pentose phosphate, complement-coagulation cascade (at 24 h for the last three), RIG-I signaling receptor (Figure 2C), TGFβ signaling, and IFNα signaling (at 6 and 24 h) (Figure 2D). A number of pathways were downregulated after MOPV infection: ribosomes (at 6 and 24 h), TGFβ signaling, antigen processing/presentation, ubiquitin mediated proteolysis, TLR receptor signaling (Figure 2C), IFNα signaling (at 48 h), IL-6 JAK-STAT3 signaling, complement, IFNγ signaling, and TNFα signaling via NFκB (at 48 h for the last two) (Figure 2D). Relative to MOPV infection, LASV infection activated the following pathways: ribosomes, NOTCH signaling, JAK-STAT signaling, apoptosis, NOD-like, TLR, and RIG-I like receptor signaling, ubiquitin-mediated proteolysis (Figure 2C), TGFβ signaling, IL-2 STAT5 signaling, ROS pathway, IL-6 JAK-STAT3 signaling, IFNγ signaling, and complement (Figure 2D). Finally, relative to LASV infection, MOPV infection activated the following pathways: oxidative phosphorylation, cell-adhesion molecules, complement-coagulation cascade (Figure 2C), protein secretion, and the inflammatory response (Figure 2D).

We more finely characterized the response of HUVECs to LASV and MOPV infection by analyzing the expression of genes representative of pathways relevant to the biology of ECs [24,38,39]. Although LASV infection led to moderate upregulation of genes involved in the type I IFN response 24 and 48 h after infection, MOPV induced robust activation of these genes after 48 h (Figure 3). The responses to LASV and MOPV were strikingly different for RLR, TLR, and NFκB signaling-related genes, with early and intense upregulation of a number of genes with LASV, whereas the upregulation of gene expression was observed 48 h after infection with MOPV and concerned other genes. We observed a similar pattern for genes related to cytokines, TNF signaling, antigen processing and presentation, and cell adhesion molecules (Figure 4 and Appendix A), with early expression of these gene sets after LASV infection and late expression after MOPV infection. Among the genes induced by LASV 6 h after infection were those for cytokines (IL-11, IL-6, TGFβ), cytokine/chemokine receptors (IL-6R, IL-7R,IL-1RAP, IL-20RB, IFNGR2, IL-6ST, CCR10, CCR4, TNFRSF, CSF2RB, NGFR), an activation receptor (CD40, Fas), transcription factor-related genes (NFKB1, ATF2, RELA, CEBPB, NFYA, RFX5, RFXANK), cAMP response element-binding proteins (CREB1, CREB3L4, CREB5), apoptosis-related genes (CASP3, CASP8, BIRC2, CFLAR, TRADD), extracellular matrix interaction-related proteins (MMP14), kinases (AKT2, MAPK12, MAPK13, RIPK3, MAP2K6, MAP3K8, RIPK1), chaperone proteins (HSP90AB1, CANX, CALR), and cell-adhesion molecules (NECTIN1, NECTIN2, ICAM2, ICAM3). The genes induced by MOPV 48 h after infection included those for cytokines/chemokines (IL-7; IL-15; IFNβ1; IL-12A; CCL1, 2, 7, 8, and 17; CXCL2, 8, 9, 10, 11, and 16; CX3CL1; CSF1), cytokine/chemokine receptors (IFNλR1, IL12-RB1, CCR4, TNFRSF, OSMR, LIFR), transcription factor-related proteins (NFKBIA, JUNB, TNFAIP3), apoptosis-related proteins (CASP7, 10, BIRC3, TRADD, CFLAR), kinases (RIP1K, MAP2K6, MAP3K8, MAPK10, MLKL), HLA and antigen processing-related proteins (HLA-A, B, C, E, F, and G; HLA-DPA1 and DPB1; DOB; B2M; TAP1; TAPBP; PSME1 and 2; CTSS; CIITA), and cell-adhesion molecules (ICAM1, VCAM1).

### 3.3. Different Proteomic Profiles Are Induced in HUVECs by LASV and MOPV

We next explored whether the differences in gene expression induced by the two viruses led to differences in protein synthesis. First, we quantified 72 soluble proteins in the cell supernatants using Luminex technology (see the Materials and Methods section for the entire list of analytes). The quantity of only five differed according to the condition (Figure 5). We detected elevated levels of CXCL10, although not significant, in MOPV-infected HUVEC supernatants 48 h after infection. We observed a highly similar increase in IL-6 concentrations in the supernatants of LASV- and MOPV-infected cells 48 h after infection, but the change was only significant for MOPV. A low but significant increase in granzyme B (GrzB) release was measured 48 h after MOPV infection, whereas a nonsignificant increase of endoglin levels was observed at the same timepoint for both viruses. Finally, there was a nonsignificant increase in thrombospondin-2 levels 48 h after infection with LASV, but not MOPV.

We performed a label-free shotgun proteomics analysis with high-resolution tandem mass spectrometry on cell extracts and cell supernatants harvested 18 and 48 h after infection to obtain a deeper insight into the proteomic changes (Figure 6). As the MOPV and LASV infections were not performed together, it was only possible to compared infected cells with their mock counterparts but not LASV- with MOPV-infected cells. Only a few proteins were found to be differentially secreted between mock and infected cell supernatants (Figure 6A). There was an increase in MAST4, FABP1, and KLKB1 levels that lasted until 18 h after LASV infection, whereas increased concentrations of IGFBP3 lasted until 48 h after infection. On the contrary, MOPV infection led to the late (48 h) release of PSMD2, RPS11, HNRNPA3, ATP1A1, TARS1, and PRKAR1A, whereas significantly elevated levels of IGFBP3 were found at both timepoints. Only a few differentially abundant (DA) proteins were found after LASV infection in HUVEC extracts harvested 18 and 48 h after infection (Figure 6B). The few proteins for which a fold-change >2 or <−2 was observed 48 h after infection are involved in apoptosis (SCARF1, PLSCR3), ubiquitination (UBE2E3, OTULIN), or the response to type I IFN (OAS2). By contrast, although only four DA proteins were detected 18 h after MOPV infection in the cell extracts, and 61 proteins were DA 48 h after infection (Figure 6C). However, a fold-change >2 or <−2 was only found for 26 and 9 proteins, respectively. These proteins belong to the type I IFN response family (DDX60, IFI44, IFIT2, 3, MX2, OAS1, 2, OASL, ISG20, PARP12, HERC6, UBE2L6, EPSTI1, SAMD9, USP18, and CMPK2), NFκB pathway (PAZRP10), ubiquitination pathway (RBCR1, UBE2E3), or apoptosis pathway (PLSCR3, RIPK1), or are transcription factors (SP110, HELZ2). The other DA proteins did not show substantial fold-changes after infection. They were mostly type IFN response-related proteins, and some belonged to the antigen presentation and processing pathway.

We also compared the proteome of HUVECs according to several pathways relevant to EC biology. The results confirmed that MOPV, but not LASV, induces a robust type I IFN response 48 h after infection (Figure 7). Neither TNF nor TLR signaling pathways were stimulated by either virus. Several proteins involved in the RIG-I like receptor signaling pathway were upregulated 48 h after infection of HUVECs with MOPV. Cytokine/cytokine receptor, cell adhesion molecule, and NFkB signaling pathways were not altered after LASV or MOPV infection. Finally, among the proteins involved in the antigen processing/presentation pathway, only HLA-A, HLA-B, and TAPBP were significantly upregulated 48 h after MOPV infection.

## 4. Discussion

ECs are pivotal in the pathogenesis of viral hemorrhagic fevers, including LF. Indeed, although coagulopathy and thrombocytopenia contribute to the hemorrhagic signs and hypovolemic and hypotensive shock observed during severe LF [40,41], vascular leakage due to increased endothelial permeability also plays an important role in these events [26]. Increased EC permeability and the disruption of adherens junctions induced by Old and New World arenaviruses have been demonstrated in vitro [42,43] and their involvement in the pathogenic cascade confirmed in animal models of arenavirus diseases [44,45,46]. ECs are a privileged target for LASV, as demonstrated in vitro in HUVECs [23] and in NHP models [22,47]. We further investigated the mechanisms by which ECs participate in LASV pathogenesis by comparing infection with LASV to that with MOPV, a nonpathogenic arenavirus closely related to LASV [16,18,48]. A similar study was performed by Lukashevich et al. in 1999 [23], but the use of transcriptomic and proteomic approaches not available at that time should make it possible to obtain new insights into EC biology during arenavirus infection. Here, we confirm that these cells are indeed permissive to both LASV and MOPV infections, with high viral titers released as soon as two days post infection (DPI) and lasting up to 4 DPI. Infection of HUVECs with both viruses is not cytopathic, as the cell viability remained similar between virus- and mock-infected cells. This observation suggests that a bias in transcriptomic and proteomic responses between infected and control cells because of dead cells or debris is unlikely. Productive infection of ECs could explain the pantropic dissemination that is associated with fatal outcomes [24] by allowing LASV to reach all organs and tissues. Consistent with such a mechanism, the detection of LASV antigens within ECs is a late event during the course of the disease in cynomolgus monkeys, and it is observed at a time when LASV has spread throughout the body [22,24]. As previously reported [23], we observed higher viral titers in supernatants of LASV-infected HUVECs than in those of MOPV-infected cells. However, the difference was only significant during the first two days after infection, suggesting that the ability of the two viruses to replicate in ECs is not central to their difference in pathogenicity.

We finely characterized the response of HUVECs to both viral infections by analyzing their transcriptome and proteome. LASV stimulated greater synthesis of mRNA than MOPV by cells early after infection and up to 24 h. The transcriptomic profile suggests that RLR-, TLR-, NFκB-, and TNF-signaling pathways were activated 6 h after LASV infection. A number of genes involved in antigen-processing/presentation and cell adhesion was also upregulated 6 h post infection. These results indicate that cell sensors are rapidly triggered after LASV infection, which does not appear to be the case with MOPV. However, this early stimulation was followed by only the modest upregulation of genes and proteins belonging to the type I IFN response in LASV-infected HUVECs, and no further synthesis of mRNA coding for cytokines and chemokines or proteins involved in antigen processing/presentation or cell adhesion was observed from 24 h after infection. We obtained similar results for the corresponding proteins. Such a rapid shutdown of the transcription of genes involved in innate immunity and antiviral responses following LASV infection is consistent with the immunosuppressive exonuclease domain contained in the LASV NP. Indeed, arenavirus NP includes an exonuclease capable of digesting double stranded (ds) RNA, resulting in the absence of cell sensing of the viral infection and type I IFN synthesis [49,50,51]. The early transcriptional response observed 6 h after LASV infection may result from viral material present in the infecting particles. As soon as new NP is produced within HUVECs, its immunosuppressive effect may shut down any further innate response. In contrast to LASV infection, no significant gene or protein synthesis was observed before 48 h after MOPV infection. The lack of an early antiviral response relative to LASV infection is unclear, as MOPV is known to induce a more intense type I IFN response than LASV [19,20,23,52]. Differences between MOPV and LASV particle entry processes may result in differences in the stimulatory capacity of viral material released within the cytoplasm. However, HUVECs were strongly activated by 48 h after MOPV infection, as illustrated by the robust synthesis of genes involved in innate immunity, cytokine/chemokine signaling, and antigen processing/presentation. In particular, IFN-stimulated genes were significantly transcribed 48 h after MOPV infection, and the corresponding proteins were detected in HUVECs. Why a robust type I IFN response was induced at this timepoint even though the NP of MOPV contains exonuclease activity similar to that of LASV NP is unclear [53] but is consistent with the significant type I IFN response observed in other cell types after MOPV infection [20,21,54,55]. The strong synthesis of mRNA coding for chemokines, cytokines, and proteins involved in antigen processing/presentation by MOPV-infected HUVECs suggests that ECs may be able to induce adaptive immune responses in vivo and confirms the higher immunogenicity of MOPV relative to that of LASV, as previously observed in vitro and in NHP models [18,21,23,48,54,55,56]. Moreover, elevated levels of CXCL10, a chemokine known to attract monocytes, T cells, NK cells, and dendritic cells and to promote T-cell adhesion to ECs [57,58,59], were detected in MOPV-infected HUVEC supernatants but not those of LASV-infected cells. Interestingly, low levels of circulating CXCL10 have been associated with fatal LF in humans [60]. As ECs are important nonprofessional antigen-presenting cells [61], the lack of activation of LASV-infected ECs may play a role in the deficient specific T-cell responses involved in severe LF [24,62]. Moderate levels of IL-6 were found in supernatants of infected HUVECs, which is reminiscent of the elevated concentrations of IL-6 associated with severe LF [24,62,63].

Aside from proteins involved in innate immunity, we observed other differences between LASV- and MOPV-infected HUVECs. Low levels of GrzB were released into the supernatants of MOPV-infected HUVECs. This mediator could act on endothelial permeability, as GrzB is known to release vascular endothelial growth factor from the extracellular matrix, resulting in vascular permeability [64]. However, the production of GrzB by ECs during LF is unlikely, as it was not detected in LASV-infected HUVEC cultures, and only minute amounts were measured in MOPV-infected cells. Furthermore, the production of GrzB appears to be restricted to immune cells. Moderate concentrations of thrombospondin-2 (TSP2) were released into the supernatants of LASV-infected HUVECs. TSP2 belongs to a family of matricellular proteins involved in the regulation of cell-matrix interactions and is known to inhibit angiogenesis through a direct effect on EC migration, proliferation, and apoptosis and by antagonizing VEGF activity [65]. However, TSP2 has also been shown to have anti-inflammatory properties through the activation of regulatory T cells and M2 macrophages and to decrease vascular permeability induced by LPS treatment [66]. Thus, TSP2 may be expressed to counterbalance the early inflammatory response induced by LASV in HUVECs. Levels of soluble endoglin (sENG) were also slightly elevated 48 h after infection for both viruses, perhaps because of the activation induced by viral infection, as previously observed with dengue virus [67]. sENG has been shown to induce a pro-inflammatory phenotype in ECs through the activation of the NFκB/IL-6 pathways [68]. However, the endothelial permeability modulated by sENG can be increased or decreased, depending on the model studied [69,70].

Only a few proteins were found to be differentially secreted into the supernatants of infected cells, but striking differences were observed between the two viruses. Indeed, LASV induced the early release of MAST4, a serine/threonine kinase [71]; FABP1, a lipid chaperone [72]; and kallikrein (KLKB1). Tissue kallikrein, a serine protease, is known to be synthesized and released by human ECs, resulting in the generation of kinins, such as bradykinin [73]. Bradykinin is a potent proinflammatory peptide that has several effects on the endothelium, including vasodilatation and increased vascular permeability [74]. Therefore, the activation of the kallikrein–kinin system by LASV, but not MOPV, could play a role in the clinical signs observed during LF. Consistent with such a role, activation of the kallikrein–kinin system has been shown to be involved in EC permeability during hantavirus infection [75], and blockade of this pathway in patients with COVID-19 helps to prevent acute respiratory distress [76]. By contrast, a number of proteins was detected at later stages in MOPV-infected HUVECs, such as PSMD2, a component of the ubiquitin-proteasome pathway; RPS11, a ribosomal protein; HNRNPA3, a protein involved in RNA processing and splicing; ATP1A1, the alpha subunit of a Na+/K+ pump; TARS1, the threonyl-tRNA synthetase; and PRKAR1A, the type 1A regulatory subunit of protein kinase A. However, the relevance of the presence of these proteins in MOPV-infected cell supernatants is unclear. We also observed striking differences in the intracellular protein content according to the virus. Indeed, although only a few proteins were DA after LASV infection, MOPV induced the expression of numerous proteins 48 h post infection. Among the proteins upregulated by LASV were a chaperon (HSPA6), a kinase (PHKG2), a deubiquitinase (OTULIN), an IFN-induced protein (OAS2), and a protein involved in microtubule dynamics (SLAIN2). A poly(ADP-ribose) polymerase (PARP9) known to serve as a noncanonical sensor for RNA viruses was also expressed [77]. A few proteins were downregulated after LASV infection relative to the mock condition: transcription factors (MAFF, BTF3L4) and proteins involved in apoptosis and the clearance of apoptotic cells (PLSCR3, SCARF1) [78,79], ubiquitination (UBE2E3), and mTOR (RRAGA) [80] and NOTCH signaling (POGLUT2) [81]. In contrast to LASV infection, only a single protein (CLPP) was overexpressed early after MOPV infection, and three were downregulated. The downregulated proteins were a kinase (SRC), a protein known to protect against oxidative stress (LanCL1) [82], and POGLUT2. A number of proteins downregulated 48 h after infection were similar between LASV and MOPV (UBE2E3, PLSCR3) or specifically modulated during MOPV infection, such as an RNA-binding protein (FRX2), a mediator involved in RNA metabolism (SREK1), a proteasome-interacting molecule (PITHD1) [83], TMEM109, a subunit of the RNA exosome complex (EXOSC7), and a deubiquitinase (USP13). In contrast to LASV infection, numerous intracellular proteins were present 48 h after MOPV infection. Most belong to the IFN-response family (ISG15/20, IFI16/35/44, IFIT2/3, MX2, OAS1/2/L, UBE23L6, SAMD9, HERC6, EPSTI1, DDX58/60, PARP9/12, CMPK2, USP18, STAT1/2, ADAR, RTF2), but proteins involved in antigen processing/presentation (TAPBP, HLA-A), apoptosis pathway (RIPK1), ubiquitination (RBCK1), and transcription (SP110, HELZ2, PHAX) were also expressed 48 h after infection. Transcriptomic and proteomic analyses gave for the most part consistent results, with DE genes leading to DA proteins in the same culture conditions. This was the case for the type I IFN response, but also for RIPK1, STAT1, DDX58, ISG15, TRIM25, and MYD88, involved in TNF-, TLR-, RLR-, and NFκB signaling, 48 h after MOPV infection. Similarly, HLA-A and B, involved in cell adhesion and antigen presentation pathways, were DE and DA in cell extracts at the mRNA and protein levels, respectively, 48 h after infection with MOPV, and to a lesser extent, LASV.

Although both LASV and MOPV replicate at substantial levels in HUVECs, the cellular response is dramatically different, with an early moderate response that is rapidly shut down after LASV infection versus a late but more intense response after MOPV infection. The immunosuppressive exonuclease activity encoded in the NP may be responsible for the rapid extinction of the HUVEC response after LASV infection, despite robust and lasting replication. The lower type I IFN response observed at the mRNA and protein levels 48 h after infection with LASV compared to MOPV is consistent with this hypothesis and could explain the rapid shutdown of the other pathways that are mostly linked to the type I IFN response. The robust responses induced at late stages of MOPV infection include the type I IFN response and antigen processing/presentation pathways. This is consistent with the higher immunogenicity of MOPV relative to that of LASV observed in other in vitro models [20,21,54,55]. The reason for the discrepancy between LASV and MOPV, despite apparently equivalent exonuclease activity, is still unclear [53]. One possibility is that a larger amount of pathogen-associated molecular patterns was induced during MOPV infection than during LASV infection, thus overloading the exonuclease activity. In conclusion, this study shows that both LASV and MOPV productively infect HUVECs, which is consistent with the privileged tropism of LASV for ECs during LF. However, MOPV induces much stronger activation of HUVECs than LASV, for which a rapid shut down of cell activation occurs.

## Figures and Tables

**Figure 1 viruses-14-00652-f001:**
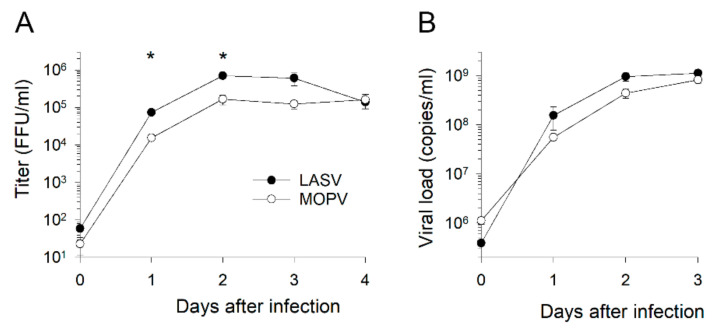
LASV and MOPV replication in HUVECs. (**A**) Viral titers, expressed in FFU/mL, and (**B**) viral RNA loads, expressed in copies/mL, are presented according to the number of days after infection with LASV (black circles) or MOPV (white circles). Results are expressed as the mean ± standard error of the mean (SEM) from three different cell donors. Asterisks represent significant differences (*p* < 0.05) between LASV and MOPV titers by a *t*-test.

**Figure 2 viruses-14-00652-f002:**
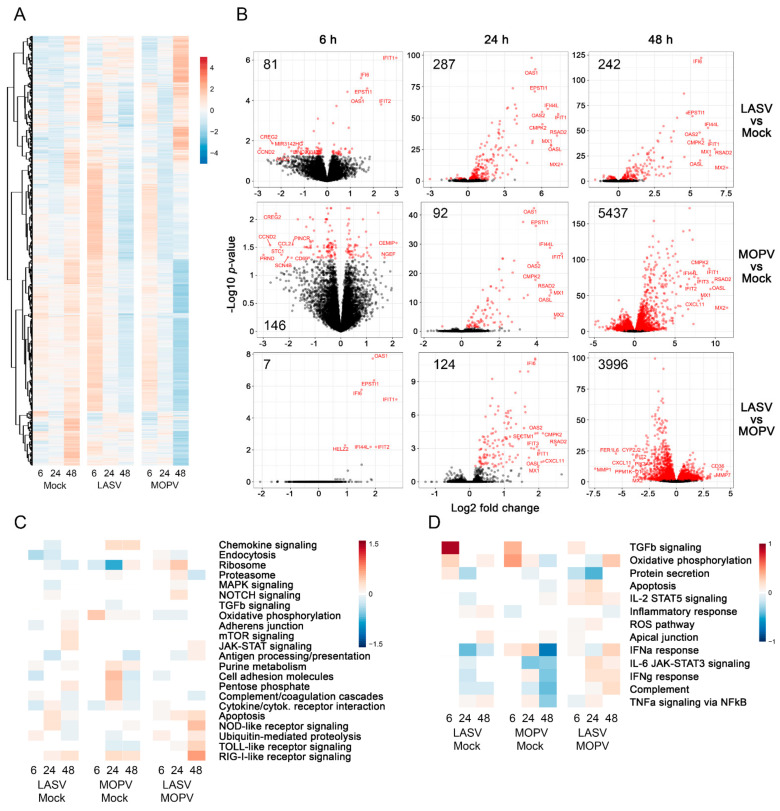
Transcriptomic profiles of infected HUVECs. (**A**) Heatmap of the total DE genes (absolute Log2 fold-changes) 6, 24, and 48 h after infection. Gene expression was standardized by VST transformation; hence, the results are centered and scaled to make the gene expression comparable. Each column represents the mean gene expression of the three donors of each group for a given timepoint. (**B**) Volcano plots representing the DE genes (red circles) (threshold of adjusted *p*-value < 0.05) between LASV and mock (upper row), MOPV and mock (middle row), and LASV and MOPV conditions (lower row) at 6 (left column), 24 (middle column), and 48 h (right columns) after infection. Each circle corresponds to the mean gene expression of the three donors. The *x*-axis reflects the Log2 fold-changes in expression for each given gene in the first condition with respect to the second condition. The *y*-axis accounts for the statistical significance of these results calculated as the–(minus) Log10 of the *p*-values. The number of DE genes for each comparison is shown in the upper left of graphs. Heatmaps of the CAMERA scores for the most significant pathways according to the analysis performed on KEGG (**C**) and Hallmark (**D**) gene sets from MSigDB. Comparisons between groups are shown on the *x*-axis and gene sets on the *y*-axis. The color of each cell (red and blue) corresponds to pathways that were significantly upregulated or downregulated, respectively. A gene set is considered to be upregulated or downregulated if its CAMERA score is positive or negative, respectively.

**Figure 3 viruses-14-00652-f003:**
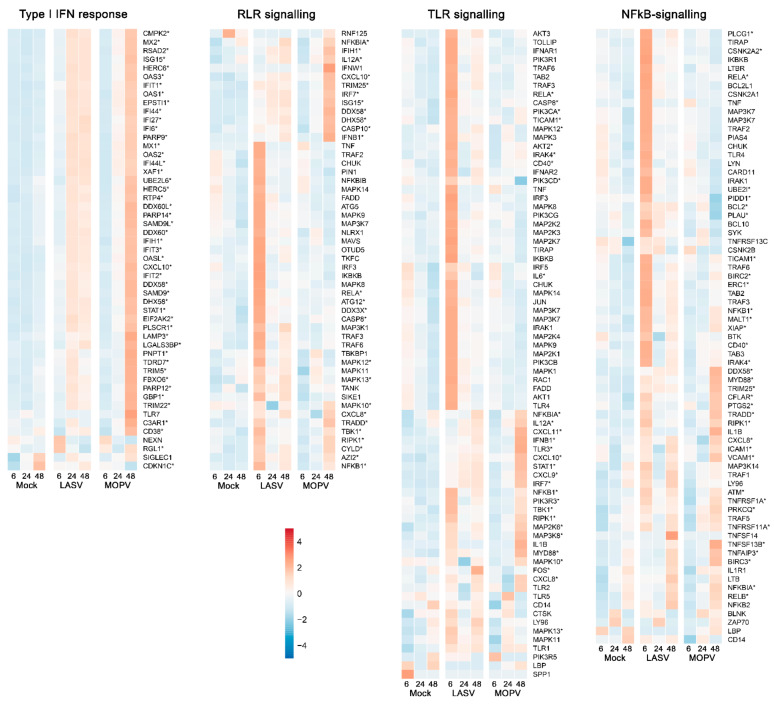
Heatmaps of the gene expression profiles of gene sets related to the type I IFN response and RLR, TLR, and NFκB signaling. Genes that are DE in any of the pairwise comparisons are highlighted with asterisks. Gene expression was standardized by VST transformation, centered, and scaled to make the gene expression comparable, and hence averaged by condition and timepoint. *p*-values were adjusted for multiple-testing using the Benjamini–Hochberg correction.

**Figure 4 viruses-14-00652-f004:**
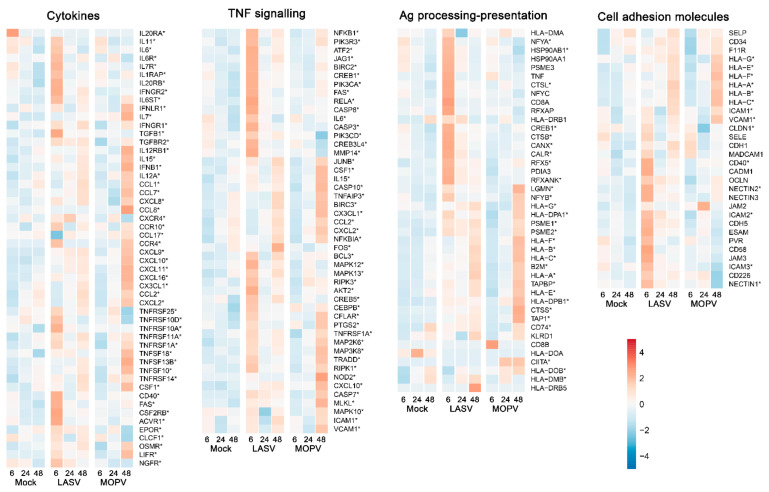
Heatmaps of the gene expression profiles of gene sets related to cytokines, TNF signaling, antigen processing/presentation, and cell adhesion molecules. Results are shown as in Figure 3. Genes that are DE in any of the pairwise comparisons are highlighted with asterisks.

**Figure 5 viruses-14-00652-f005:**
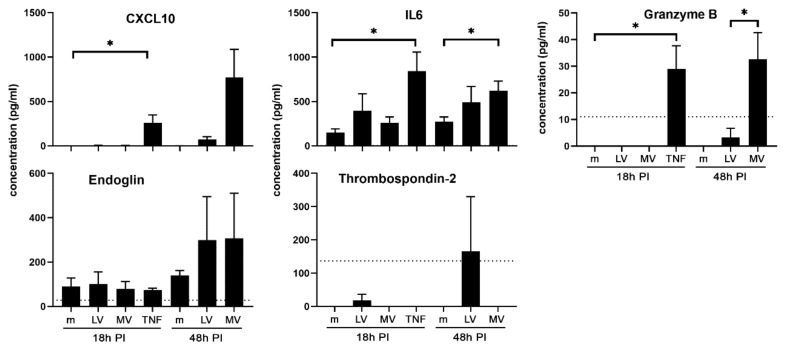
Quantification of mediators in the supernatants of infected HUVECs. Several mediators were quantified in the supernatants of uninfected HUVECS or HUVECs infected with LASV or MOPV 18 and 48 h after infection. Supernatants of TNFα-stimulated HUVECs were used as a positive control. Results are expressed in pg/mL as the mean ± SEM from three different cell donors. Asterisks represent significant differences (*p* < 0.05) between different conditions by a *t*-test. The dotted lines indicate the limit of detection of the assays.

**Figure 6 viruses-14-00652-f006:**
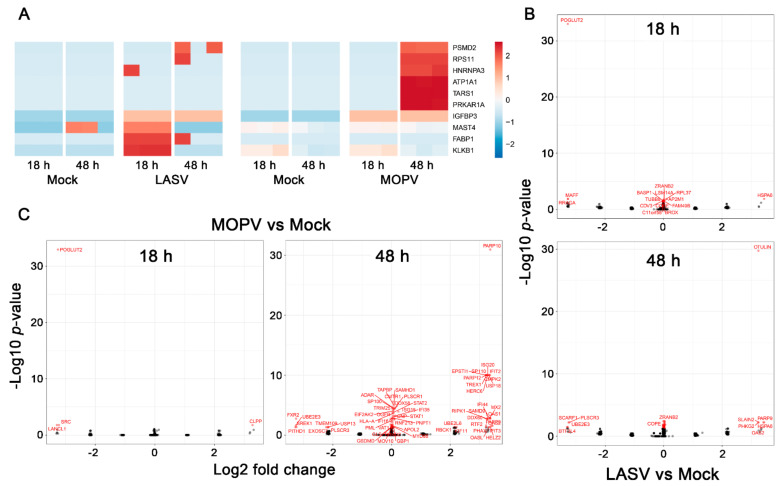
Proteomic profiles of LASV- and MOPV-infected HUVECs. (**A**) DA proteins (Log2 fold-changes) in supernatants of HUVECs according to the time after infection and between LASV or MOPV and its respective mock condition. Each timepoint represents an individual cell donor. The color indicates the average standardized (centered and scaled) protein abundance. (**B**,**C**) Volcano plots representing the DA proteins in HUVEC extracts according to the time after infection and between LASV (**B**) or MOPV (**C**) and its respective mock condition. Each circle corresponds to the mean protein expression of the three donors of each group for a given timepoint. The *x*-axis reflects the log2 fold-changes in abundance for each given protein in the first condition with respect to the second condition. The *y*-axis accounts for the statistical significance of these results calculated as the Log10 of the *p*-values. DA proteins are represented by a red circle and are identified using their gene symbol.

**Figure 7 viruses-14-00652-f007:**
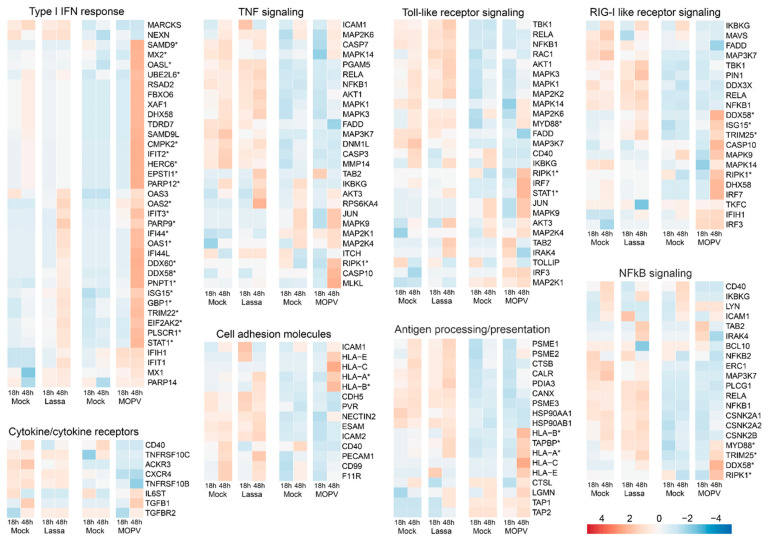
Heatmaps of protein abundance of different protein sets in HUVEC cell extracts. DA proteins are highlighted with asterisks. The average quantity is indicated on the scale.

## Data Availability

The datasets related to RNA sequencing and proteomics generated during and/or analyzed during the current study are not publicly available owing to further investigation currently in progress but are available from the corresponding author upon reasonable request.

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
