# Peer review of "Infection of Human Endothelial Cells with Lassa Virus Induces Early but Transient Activation and Low Type I IFN Response Compared to the Closely-Related Nonpathogenic Mopeia Virus"

_viruses, 2022, doi:10.3390/v14030652_

Round 1
Reviewer 1 Report
General comments:
In this manuscript, Merabet and colleagues provide evidence that both Lassa virus (LASV) and nonpathogenic Mopeia virus (MOPV) productively infect human umbilical vein endothelial cells (HUVECs). HUVECs are a surrogate model system to investigate viral multiplication in vascular endothelial cells that is highly relevant to the pathogenesis of Lassa fever. The authors examined the transcriptomic and proteomic dynamics of HUVECs infected with LASV and MOPV and revealed that MOPV induced higher expression levels of genes related to innate immunity than LASV, which may contribute to the early suppression and uncontrolled dissemination of MOPV and LASV, respectively.
The paper is reasonably well written, and the experiments have been performed well. The results are clearly presented. A general limitation of this descriptive work is the lack of validation assays to examine the roles of the genes identified by the authors in viral multiplication/pathogenesis in vitro and in vivo. Regardless of this limitation, this manuscript provides key comprehensive information that is expected to contribute to future follow-up studies to elucidate the molecular mechanisms underlying the pathogenesis of LASV as well as the nonpathogenic nature of MOPV.
There are several aspects of this work that could be improved by providing additional clarification, as noted below.
Specific comments:
1) I appreciate that the authors described a large number of differentially expressed genes and their potential roles in viral multiplication and pathogenesis. However, the central message of the authors is unclear. I would suggest changing the title and shortening the Discussion section to help the readers more easily grasp the key contribution of this manuscript.
2) Does LASV or MOPV cause cytopathic effects or cell death in HUVECs? Cell debris/dead cells may affect expression profiling.
3) Lines 553–556: I disagree with the authors’ argument that “viral replication is quite similar between LASV and MOPV in HUVECs” as the LASV titers are ~10-fold higher than the MOPV titers at 1, 2, and 3 days post-inoculation (Fig. 1).
Minor points:
None of the Greek letters is correctly displayed.
Reviewer 2 Report
In this manuscript, Merabet et al compared the transcriptomic and proteomic responses of HUVEC cells to infection with the two viruses to elucidate the mechanisms involved in their differences in immunogenicity and pathogenicity.
Firstly, the authors examine the replication kinetics of LASV and MOPV in HUVEC cells over 4 days and show that LASV has a higher replication in the first 2dpis and there are some differences in the genome copies/pfu ratio. Figures 2, 3 and 4 present the results of the RNAseq analysis. Major pathways of interest are identified and then heatmaps for expression changes of associated components are discussed. Figure 5 discusses results of the Luminex assay performed on the supernatants, with five genes being differentially expressed. This is followed up by shotgun proteomic analysis of the supernatants. Figure 6 shows the proteomic profiles of LASV or MOPV infected cells compared to mock infection followed up by a more detailed analysis of proteins associated with individual pathways of interest.
Overall, this was an interesting paper and a well-organized manuscript. I have no serious concerns with the methodologies or any major issues with the manuscript in general.
However, though the authors are employing two broad based screening methodologies, they do not discuss the RNAseq and proteomic hits in relation to each other. Some genes (eg STAT1, MX2 IFIT1) seem to be differentially regulated at a transcriptional as well as translational level. Are there any differences? A few lines of discussion comparing and contrasting the two methods will strengthen applicability and overall findings of the paper.
Other Minor points:
- Mastomys natalensis needs to be italicized on Line 52 and 56.
- Line 259: (RNAseq) instead of (seq)
- I’m assuming that DE on line 261 in stands for “differentially expressed”? If so, please add the full form there for clarity.
- In figure 5: The labels of the the CXCL10 and IL-6 panels are absent. I’m assuming they are the same as the other panels but they need to be reformatted.
- All the greek letters need to be double checked for formatting.
